# Pathways to housing stability and viral suppression for people living with HIV/AIDS: Findings from the Building a Medical Home for Multiply Diagnosed HIV-positive Homeless Populations initiative

Serena Rajabiun[1]◐*, Kendra Davis-Plourde[2]◐, Melinda Tinsley[3]‡, Emily K. Quinn[4]‡, Deborah Borne[5]‡, Manisha H. Maskay[6]‡, Thomas P. Giordano[7]‡, Howard J. Cabral[2]‡

1 Department of Public Health, Zuckerberg College of Health Sciences, Univeristy of Massachusetts, Lowell, Lowell, MA, United States of America, 2 Department of Biostatistics, Boston University School of Public Health, Boston, Massachusetts, United States of America, 3 U.S. Department of Health and Human Services, Health Resources and Services Administration, Rockville, MD, United States of America, 4 Biostatistics and Epidemiology Data Analytics Center, Boston University School of Public Health, Boston, Massachusetts, United States of America, 5 San Francisco Department of Public Health, San Francisco, CA, United States of America, 6 Prism Health North Texas, Dallas, Texas, United States of America, 7 Department of Medicine, Baylor College of Medicine, The Center for Innovations in Quality, Effectiveness and Safety (IQuESt), Michael E. DeBakey VA Medical Center, and Thomas Street Health Center, Harris Health System, Houston, Texas, United States of America

◐ These authors contributed equally to this work.
‡ These authors also contributed equally to this work.
* Serena_Rajabiun@uml.edu

**Data Availability Statement:** The data underlying the results presented in the study are available

## Abstract

### Background

People with HIV with co-occurring substance use and mental health diagnoses who are unstably housed have poorer outcomes for retention in care and viral suppression. Navigation models are a potential strategy to help this vulnerable population obtain the necessary medical and non-medical services across multiple service systems. The Health Resources and Services Administration's Special Projects of National Significance: "Building a Medical Home for Multiply-Diagnosed HIV-positive Homeless Populations initiative 2012–2017 found that navigation models may be an effective intervention to support people with HIV with unstable housing improve HIV health outcomes. However, there is limited information about the mechanisms by which this intervention works. In this article, we explore the participant and program factors for achieving stable housing at 6 months and how these factors influence HIV health outcomes.

### Methods and findings

This was a prospective study of 471 unstably housed people with HIV enrolled in a navigation intervention across nine sites in the United Stated from 2013–2017. All sites provided HIV primary medical care. Eight sites were located in urban areas and one site served a

upon request from the Biostatistics and Epidemiology Data Coordinating Center. Due to the sensitive nature of the data and as required by the Boston University Medical Campus Institutional Review Board (IRB). datasets are stored for up to seven years from the close of the study, and thus with this project through 2025. Data requests can be sent to Boston University's Biostastics & Epidemiology Data Analytics Center (BEDAC) @ bedacprp@bu.edu. This request will be sent to the study's Publications & Dissemination Committee, which consists of the Principal Investigators from each local study sites, the multisite evaluation center at Boston University and HRSA. As Principal Investigator for the multisite evaluation center I will manage the approval process, in accordance with our policy and guidelines Once a data request is approved by the P & D committee, BEDAC will work with the requesting party to create a dataset for processing fee.

**Funding:** This study was funded by U.S. Department of Health and Human Services, Health Resources and Services Administration under grant #U90HA24974.

**Competing interests:** No authors have competing interests

predominantly rural population. Two sites were federally qualified health centers, three were city or county health departments, one site was a comprehensive HIV/AIDS service organization, and three sites were outpatient or mobile clinics affiliated with a university -based or hospital system. Data were collected via interview and medical chart review at baseline, post 6 and 12 months. Type and dose of navigation activities were collected via a standardized encounter form. We used a path analysis model with housing stability at 6 months as the mediator to examine the direct and indirect effects of participant's socio-demographics and risk factors and navigation on viral suppression and retention in care at 12 months. Housing stability at 6 months was associated with male gender, younger age, viral suppression at baseline, having a lower risk for opiate use, recent homelessness, lower risk of food insecurity, and a longer length of time living with HIV. Participants who increased self-efficacy with obtaining help by 6 months had significantly higher odds of achieving housing stability. Stable housing, fewer unmet needs, moderate to high risk for opiate use, and viral suppression at baseline had a direct effect on viral suppression at 12 months. The intensity of navigation contact had no direct effect on housing stability and a mixed direct effect on viral suppression. Recent diagnosis with HIV, women, greater social support, increased self-efficacy and higher intensity of navigation contact had a direct effect on improved retention in HIV primary care at 12 months.

## Conclusions

In this sample of people with HIV who are experiencing homelessness, housing stability had a significant direct path to viral suppression. Navigation activities did not have a direct effect on the path to housing stability but were directly related to retention in care. These results identify key populations and factors to target resources and policies for addressing the health and social unmet needs of people with HIV to achieve housing stability and HIV health outcomes.

## Introduction

For people with HIV having stable, secure, and adequate housing is a significant factor in obtaining appropriate HIV medical care, access and adherence to antiretroviral therapy (ART), achieving viral suppression, and reducing risk of transmission [1–7]. Even among people with HIV who have access to medical care, housing stability remains a challenge to reaching viral suppression. According to national data from the Ryan White HIV/AIDS Program (RWHAP), the payor of last resort for HIV medical care in the US, 86% of participants with stable housing reached viral suppression compared to 72% of participants who were unstably housed [8]. Interventions that support people with HIV with obtaining stable housing and other social and medical needs are needed for this vulnerable population group.

The Health Resources and Services Administration's Special Projects of National Significance aimed to address this disparity through the Building a Medical Home for Multiply-Diagnosed HIV-positive Homeless Populations initiative from 2012–2017 (HRSA SPNS Homeless Initiative). Patient navigation models are a potential strategy to help people with HIV who experience homelessness obtain the necessary medical and non-medical services necessary by coordinating and accessing care across multiple services systems [9, 10]. Patient navigators are members of the care team who can provide this intensive service and work with medical and

behavior health care providers to provide a seamless system of care. Results from this initiative demonstrated that navigation models were an effective intervention to support people with HIV who experience homelessness achieve more stable housing, improve retention in care, and reach viral suppression [6, 11]. Approximately 60% of SPNS participants were able to achieve temporary or permanent supportive housing. Among those who stabilized their housing, 86% were retained in appropriate HIV care and 77% achieved viral suppression compared to those who remained unstably housed (79% were retained in care and 66% were virally suppressed) in the post intervention period [6].

In addition, the initiative found that for people with HIV experiencing homelessness, the road to continuous housing stability is achievable with sufficient support. Approximately 43% of participants were able to obtain and maintain consistent stable housing up to 12 months post intervention. This finding was statistically significant for persons with mental health disorders [AOR = 1.55; 95% CI = 1.02,2.35; p<0.05] and a history of trauma disorders [AOR = 1.72; 95% CI = 1.22,2.41; p<0.05]. However, persons with recent injection drug use had less consistent housing stability [AOR = 0.41; 95% CI = 019,0.90, p<0.05] [11]. These findings were consistent with other studies, concluding that transitions from homelessness to more stable housing were associated with a reduction in alcohol and illicit drug use and improved mental health status among those exiting incarceration and participating in a care coordination intervention [12].

Despite the promise of navigation models, there is limited information about the mechanisms through which they work to enable people with HIV experiencing homelessness achieve housing stability and improve health outcomes. This study examines the mediating effects of housing stability on HIV health outcomes and the role of patient navigation. We hypothesized that more intensive patient navigation interventions in the first 6 months would lead to housing stability at 6 months post intervention, which in turn would improve retention in HIV medical care and viral suppression at 12 months (Fig 1).

## Materials and methods

### Study design and intervention

The HRSA SPNS Homeless Initiative (2012–2017), funded nine intervention sites and one multisite evaluation center to implement and evaluate the effect of patient-centered medical homes (PCMH) on HIV health outcomes among people with HIV experiencing homelessness.

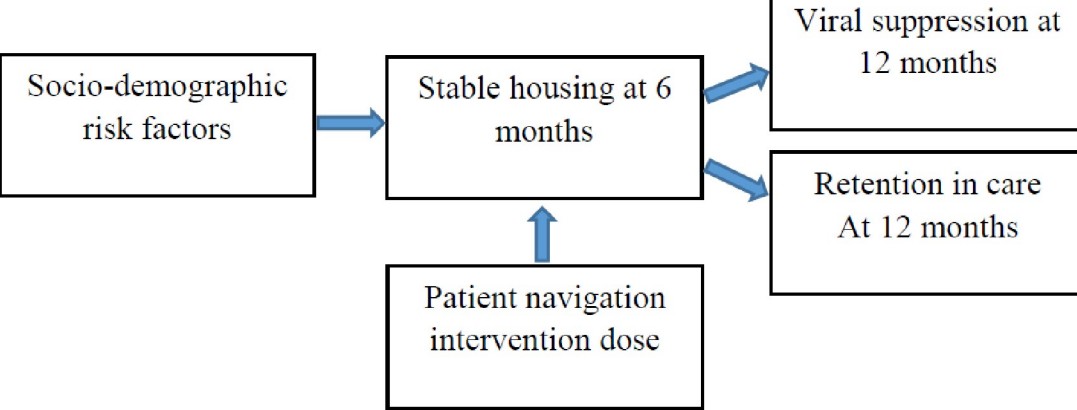

**Fig 1. Hypothesized model for pathways from homelessness to housing stability to viral suppression and retention in care for people with HIV.**

All nine sites received Ryan White funds to provide HIV primary medical care at the time of enrollment. Eight sites were located in metropolitan areas and one site served a predominantly rural population. Two sites were federally qualified health centers, three were city or county health departments, one site was a comprehensive HIV/AIDS service organization, and three sites were outpatient or mobile clinics affiliated with a university -based or hospital systems. Common elements for a PCMH across the nine sites included: 1) the use of patient navigators to conduct outreach and provide intensive individual and system coordination to address housing needs and support linkage and retention in HIV medical care; 2) the integration of behavioral health services into HIV primary care; and 3) partnerships with housing providers to obtain housing and housing assistance. Navigators were members of the care team and included peer (people with HIV) and non-peer staff. All navigators were trained in principles of harm reduction, trauma informed care and motivational interviewing techniques. They were distinct from HIV case managers and worked closely with behavioral health HIV medical, and housing providers. Housing assistance included housing search to find a place to live; assist with housing applications with agencies; linkage and coordination with Housing for Opportunities for Persons with HIV/AIDS (HOPWA) and other U.S. Department of Housing and Urban Development (HUD) resources for rental subsidies and housing units; communication and support with landlords; provision of emergency housing stays at hotels or motels for shelter resistant clients; support with finding resources to move in and furnish apartments; and access to transitional living facilities such as residential treatment for substance use disorders. Further details of the intervention have been published elsewhere [6, 9].

The HRSA/SPNS Homeless Initiative enrolled 909 participants across the nine sites. This study included a subsample of 471 SPNS participants with complete available data on our mediator (i.e., stably housed at 6 months) and at least one outcome variable of interest (i.e., retention in care at 12 months and viral suppression at 12 months). Participants gave consent and were enrolled and followed up to 12 months post intervention in a prospective, nonrandomized study across the nine sites from September 2013 through February 2017. Eligibility criteria included people with HIV who (1) were 18 years or older; (2) had a history of or current diagnosis of a substance use or mental health disorder; and (3) were currently homeless or unstably housed as defined by the U.S. Department of Housing and Urban Development (HUD) [13] *Literally homeless*: lacks a fixed, regular, and adequate nighttime residence; *Unstably housed*: an individual who has not had a lease, ownership interest, or occupancy agreement in permanent and stable housing with appropriate utilities (e.g. running water, electricity) in the last 60 days; or has experienced persistent housing instability as measured by two moves or more during the preceding 60 days (couch surfing) *and* can be expected to continue in such status for an extended period of time, *or individuals fleeing domestic violence*.

Data were collected from participant interviews and medical chart review on socio-demographic factors (e.g., gender, age, race/ethnicity, and education), housing status, incarceration history, mental health diagnoses, substance use risk factors, social support, self-efficacy for getting information, obtaining help and communicating with a physician, and unmet need for services. In addition, participants were assessed on barriers to obtaining HIV primary care including personal, organizational, and structural, as well as physical and mental health-related quality of life via interviews at baseline, and at 6- and 12-months post intervention. Further details of the study intervention, design, and measures are published elsewhere [6].

All study procedures were approved by local Institutional Review Boards at the nine participating study sites (Chesapeake IRB (PrismHealth NT and Commwell Health), San Diego State University (Family Health Centers), Baylor College of Medicine (Harris Health System), Public Health Division/Multnomah County Health Department Institutional Review Board, County of Los Angeles Public Health & Health Services Institutional Review Board (Pasadena

Public Health Department); Ethical & Independent Review Services (San Francisco Department of Public Health); University of Florida Institutional Review Board; and Yale University Institutional Review Board) and the multisite evaluation center at Boston University Medical Campus. The Office of Human Research Protection at the Department of Health and Human Services granted a certificate of confidentiality for the study.

## Measures

Our primary outcomes of interest were retention in HIV primary medical care and viral suppression at 12 months. We defined retention in HIV primary care as at least one visit in each of the three consecutive 4-month windows of the 12-month follow-up period [14]. Viral suppression was defined as having a final viral load test result, in the post 6 to 12-month observation period, of less than 200 copies per milliliter. Stably housed was defined as living in a rented or owned room, apartment or home paid for by self or permanent supported housing or subsidized housing through the Housing for Opportunities for Persons with HIV/AIDS (HOPWA) or other federal, state or local subsidy program. We measured housing stability at 6 months into the observation period. Participants who were unstably housed included persons living on the street, in public places, shelters, temporarily living with friends or family ("couch surfing"), or in a motel/hotel paid for by a program.

Navigation activities formed the core of the intervention and were defined as 43 activities across six domains: *health care related activities*: linking newly diagnosed to first medical appointment, accompanying to HIV medical appointment, follow up with HIV or non-HIV medical appointment; discuss medical appointments and help with obtaining medications. *Mental health (mh) or substance use (su) treatment support*: collect information about mental health or substance use treatment, accompany to appointments, referrals and assist with making appointments. *Housing related activities*: assist with housing application for rental assistance and housing units, creating a housing goal plan, accompany to housing appointments, provide assistance with maintaining housing, discuss housing needs. *Other social service or transportation assistance activities*: assist with obtaining transportation assistance, and assist with obtaining other social service appointments. *Educational and emotional support activities*: relationship building (checking in with client and providing emotional support), coaching on living skills, assist with disclosure, mentoring on provider interactions, education on treatment adherence, discuss safer sex, help reduce drug use/educate on harm reduction. *Employment-related or other practical support activities*, such as obtaining legal documents (IDs) food, clothing, job assistance, budgeting/financial planning, legal assistance and cell phones. Navigators completed forms on a daily basis for all encounters made directly with a client either face-to-face, phone or email/text exchange, or if the encounter was made with another health care, housing or other social service provider ("collateral") on behalf of a client. We defined *intervention dose* by generating quartiles of the total number of activities overall and by activity type during the first 6 months of the intervention. We then categorized the dose as "low", "moderate", "high" or "very high".

Other covariates included socio-demographics (gender, race/ethnicity, age, education) and risk factors that could affect our mediator or outcome variables. These measures included recent incarceration history in the past 12 months and lifetime trauma history, either physical injury or sexual assault. Food insecurity was assessed with a dichotomized variable whether a person had barely anything to eat in the past 30 days. Substance use risk was measured using the World Health Organization's Alcohol, Smoking and Substance Involvement Screening Test (ASSIST) and categorized as low, moderate (problem) or high (addictive) risk [15]. Depression risk was measured using the 10-item Center for Epidemiological studies

Depression Scale, with a score of 10 or greater indicating moderate to severe depression [16]. Social support was measured using a 5-item scale to measure types of support in the past 4 weeks with higher scores reflecting greater social support [17]. Self-efficacy was measured across 3 domains: ability to get information, obtain help and communicate with a health care provider. Each item was scored on a 10-item scale ranging from 1 =" not confident at all" to 10 =" Totally confident" [18]. Number of medical and non-medical service needs were counted and included: food, housing medication assistance, financial assistance, transportation, legal services, employment, mental health treatment, substance use treatment, and dental care. Unmet need for a service was also calculated from the list of reported need for services but unable to obtain during the previous 6 months. Barriers to care included person, organizational and structural barriers to obtaining HIV primary care [19]. Health related quality of life was measured using the Veterans RAND 12-item Health to assess domains of general health perception, physical functioning role limitations attributable to physical and emotional problems, bodily pain, energy, fatigue, social functioning and mental health. Each item is measured on a 5-point scale from "none of time" to "all of the time." The 12 items are summarized into a physical component summary score (PCS) and mental component summary score (MCS). The summary scores are set to a mean of 50 and standard deviation of 10 for the US general population [20].

## Statistical analysis

We conducted a series of univariate analyses for all continuous variables and categorical variables. These included analyses of counts and percentages for categorical variables and means, standard deviations, and quantiles for continuous variables. We also assessed the distribution of continuous variables for potential skewness or extreme values.

Using a path modeling framework, we then examined the mediating effects of housing stability at 6 months with the associations of baseline socio-demographic and risk factors and intervention dose on retention in HIV primary care and viral suppression at 12 months [21]. First, to develop the set of independent variables for our initial path model, we conducted bivariate analyses with housing stability at 6 months, retention in HIV primary care at 12 months, and viral load suppression at 12 months as dependent variables. In these bivariate analyses, independent variables associated with both housing stability and at least one clinical outcome at the 0.15 α-level were included in a subsequent comprehensive logistic path analysis model that we estimated and tested using Mplus version 8.1. Our analyses considered the multi-site design by employing study site as a clustering variable. We included in the path model client characteristics related to length of time chronically homeless and living with HIV since these characteristics could influence our mediator and outcomes of interests. Paths for independent (exogenous) variables with p-values less than 0.15 in tests of association with a mediator or outcome variable in this initial model were included in a subsequent comprehensive path model. A final more parsimonious path model was then computed after removing variables not associated with the mediator or either outcome with p-values greater than or equal to 0.15. We present adjusted odds ratios (AOR), 95% confidence intervals (CI), and p-values from this final path model with adjusted odds ratios and CIs for continuous independent variables computed per standard deviation.

## Results

As shown in Table 1, the majority of participants were cisgender men (75.0%), Hispanic/Latinx or African-American/Black (66.0%), aged 31–54 years (71.8%), and one-third had less than a high school education. Approximately 72% were homeless with an average of 6.1 years

**Table 1. Participants characteristics and associations with housing stability and HIV health outcomes, HRSA/SPNS Building a Medical Home for Multiply-Diagnosed HIV-positive Homeless Populations initiative from 2013–2017.**

| Baseline characteristics | Total (N = 471) N (%) | Stably Housed at 6 Months (N = 269) N (%) | Virally Suppressed at 12 Months (N = 266) N (%) | Retained in Care at 12 Months (N = 268) N (%) |
|---|---|---|---|---|
| Gender | | ** | | |
| Cisgender male | 353 (75.0) | 200 (56.8) | 197 (75.5) | 195 (55.2) |
| Cisgender female | 100 (21.2) | 64 (64.0) | 60 (72.3) | 64 (64.0) |
| Transgender or Other identified | 18 (3.8) | 5 (27.8) | 9 (64.3) | 9 (50.0) |
| Race/ethnicity | | | | |
| African-American/Black | 212 (45.0) | 125 (59.0) | 124 (73.8) | 121 (57.1) |
| Hispanic | 99 (21.0) | 55 (56.1) | 59 (77.6) | 55 (55.6) |
| White | 123 (26.1) | 74 (60.2) | 66 (74.2) | 70 (56.9) |
| Other (including multiracial) | 37 (7.9) | 15 (40.5) | 17 (68.0) | 22 (59.5) |
| Ages | | | | |
| 30 years or younger | 72 (15.3) | 48 (67.6) | 41 (70.7) | 44 (61.1) |
| 31–54 year | 338 (71.8) | 188 (55.6) | 185 (73.4) | 189 (55.9) |
| 55 years or older | 61 (13.0) | 33 (54.1) | 40 (83.3) | 35 (57.4) |
| Education | | ** | | |
| Less than high school | 151 (32.1) | 77 (51.3) | 85 (73.9) | 88 (58.3) |
| High school | 151 (32.1) | 99 (65.6) | 85 (72.7) | 93 (61.6) |
| Beyond high school | 168 (35.7) | 93 (55.4) | 95 (76.0) | 86 (51.2) |
| Housing status—baseline | | *** | | |
| Homeless | 341 (72.4) | 167 (49.1) | 179 (72.2) | 190 (55.7) |
| Controlled Environment | 41 (8.7) | 33 (80.5) | 27 (79.4) | 25 (61.0) |
| Unstably Housed | 89 (18.9) | 69 (77.5) | 60 (79.0) | 53 (59.6) |
| Recent Incarceration (past 12 months) | 124 (34.4) | 66 (53.2) | 62 (66.7) ** | 74 (59.7) |
| Trauma history, lifetime | | | | |
| Physical injury, harm | 205 (43.6) | 107 (52.5) * | 112 (73.7) | 118 (57.6) |
| Sexually assaulted | 194 (41.5) | 116 (59.8) | 110 (73.8) | 112 (57.7) |
| Mental Health Diagnosis prior to enrollment | 361 (80.8) | 201 (55.8) | 205 (74.8) | 214 (59.3) |
| Social support score, mean±SD | 11.3 ± 5.2 | 11.8 ± 5.4** | 11.6 ± 5.4 | 11.7 ± 5.5* |
| Change in social support score, mean±SD | 0.6 ± 5.8 | 0.6 ± 6.0 | 0.6 ± 6.1 | 1.1 ± 6.1** |
| Self-efficacy score, mean±SD | | | | |
| Getting information | 8.8 ± 2.2 | 8.9 ± 2.1* | 8.9 ± 2.1 | 8.8 ± 2.2 |
| Obtaining Help | 5.7 ± 2.4 | 5.9 ± 2.4* | 5.9 ± 2.4 | 5.8 ± 2.5 |
| Communicating with Physician | 8.7 ± 2.1 | 8.7 ± 2.0 | 8.8 ± 2.0* | 8.8 ± 1.9 |
| Change in score, mean±SD | | | | |
| Getting information | 0.1 ± 2.5 | 0.2 ± 2.3 | 0.1 ± 2.3 | 0.3 ± 2.4 |
| Obtaining Help | 0.6 ± 2.7 | 0.9 ± 2.6*** | 0.7 ± 2.5 | 0.9 ± 2.6** |
| Communicating with Physician | 0.3 ± 2.5 | 0.5 ± 2.3* | 0.3 ± 2.5 | 0.3 ± 2.3 |
| No health insurance | 172 (36.7) | 105 (61.1) | 100 (73.0) | 100 (58.1) |
| Food insecurity | 274 (58.2) | 137 (50.2) *** | 140 (69.3)** | 151 (55.1) |
| Food insecurity–need met by 6 months | 317 (67.5) | 207 (65.5)*** | 195 (78.3)*** | 190 (59.9)** |
| Need medication assistance | 250 (53.1) | 140 (56.2) | 138 (75.0) | 133 (53.2)* |
| Medication assistance–need met by 6 months | 271 (57.5) | 155 (57.4) | 158 (77.5)* | 160 (59.0) |
| Need mental health assistance | 277 (58.8) | 159 (57.6) | 152 (72.0) | 159 (57.4) |

(*Continued*)

**Table 1.** (Continued)

| Baseline characteristics | Total (N = 471) N (%) | Stably Housed at 6 Months (N = 269) N (%) | Virally Suppressed at 12 Months (N = 266) N (%) | Retained in Care at 12 Months (N = 268) N (%) |
|---|---|---|---|---|
| Mental health assistance–need met by 6 months | 224 (47.7) | 129 (57.9) | 127 (75.2) | 121 (54.0) |
| Need substance abuse treatment | 182 (38.6) | 102 (56.0) | 98 (73.7) | 94 (51.7)* |
| Substance abuse treatment–need met by 6 months | 342 (72.6) | 205 (60.1)** | 197 (74.1) | 197 (57.6) |
| Number of unmet needs, mean±SD | 3.4 ± 2.3 | 3.2 ± 2.2* | 3.1 ± 2.1*** | 3.3 ± 2.2 |
| Number of barriers to care, mean±SD | 3.2 ± 3.1 | 2.9 ± 2.8*** | 3.0 ± 3.1*** | 3.3 ± 3.0 |
| Moderate/severe risk for substance use | | | | |
| Alcohol | 193 (41.0) | 101 (52.3)* | 98 (68.5)** | 107 (55.4) |
| Cocaine | 223 (47.4) | 120 (53.8) | 133 (79.2)** | 127 (57.0) |
| Opioids | 99 (21.0) | 48 (48.5)** | 59 (80.8) | 57 (57.6) |
| Amphetamines | 162 (34.4) | 76 (46.9)*** | 81 (71.7) | 90 (55.6) |
| Moderate to severe depressive symptoms (%CES-D≥10) | 345 (73.3) | 197 (57.3) | 193 (74.2) | 195 (56.5) |
| Virally suppressed (HIV-1 RNA<200 copies/mL) at baseline | 229 (48.6) | 146 (64.0)*** | 154 (90.1)*** | 132 (57.6) |
| Health-related quality of life score, mean±SD | | | | |
| Physical composite score (PCS) | 37.9 ± 12.2 | 38.9 ± 11.8** | 38.0 ± 12.1 | 37.5 ± 12.5 |
| Mental composite score (MCS) | 35.8 ± 12.8 | 35.8 ± 13.0 | 36.3 ± 13.2 | 35.9 ± 12.5 |
| Time living with HIV in years, mean±SD | 11.2 ± 9.0 | 11.5 ± 8.8 | 11.4 ± 9.3 | 10.4 ± 8.9** |
| Years homeless, mean±SD | 6.1 ± 8.0 | 5.3 ± 6.9** | 6.2 ± 8.2 | 5.9 ± 8.1 |

*p<0.15

**p<0.05

***p<0.01.

Total column results are displayed as n (column percent) for categorical.

Remaining column results are displayed as n (row percent) for categorical.

of being homeless and 34.4% were incarcerated in the past 12 months. Approximately 40% of participants had trauma history due to physical or sexual assault and 80.8% had a diagnosed mental health disorder. Moderate to severe substance use risk was reported in 21% for opioid use, 34.4% for amphetamine use, 41% for alcohol use, and 47.4% for cocaine use. Approximately 36.7% participants had no health insurance. Participants reported multiple barriers to obtaining HIV care and unmet needs for services such as transportation, food, medication assistance, substance use and mental health treatment in addition to housing. The average number of unmet needs for services were 3.4 (SD = 2.3) and average number of barriers to care were 3.2 (SD = 3.1). Approximately 58.2% were food insecure, and 58.8% reported a need for mental health treatment and 38.6% for substance use treatment. With respect to health status, on average participants were living with HIV for 11 years, 48.6% were virally suppressed at baseline, and participants rated their physical health quality of life as 37.9 (SD = 12.2) and mental health-related quality of life as 35.8 (SD = 12.8), nearly 1.5 standard deviations lower than the general population.

Socio-demographics, including gender, age, length of time being homeless, length of time living with HIV, social support, self-efficacy, food insecurity, number of unmet needs, number of barriers to care, level of risk for opiate use, intervention dose activities and viral suppression at baseline were selected for the path analysis. Table 1 shows the characteristics of the sample

and factors associated with housing stability, retention in care, and viral suppression for the path model.

Table 2 describes the type and intensity of navigation activities by stable housing, viral suppression and retention in care. Overall, higher intensity (dose) of navigation activities were

**Table 2. Type and intensity of navigation activities by stable housing, viral suppression and retention in care.**

| I | Total | Stably Housed at 6 Months | Virally Suppressed at 12 Months | Retained in Care at 12 Months |
|---|---|---|---|---|
| | (N = 471) | | | |
| | N (%) | (N = 269) | (N = 266) | (N = 268) |
| | | N (%) | N (%) | N (%) |
| Dose—Overall (0–180 days) | | | | *** |
| Low (1–11 activities) | 116 (24.6) | 63 (54.3) | 61 (78.2) | 52 (44.8) |
| Moderate (12–26 activities) | 104 (22.1) | 59 (57.3) | 64 (80.0) | 57 (54.8) |
| High (27–54 activities) | 120 (25.5) | 72 (60.0) | 71 (71.7) | 82 (68.3) |
| Very High (55–244 activities) | 131 (27.8) | 75 (57.3) | 70 (69.3) | 77 (58.8) |
| Dose–Healthcare (0–180 days) | | | *** | ** |
| Low (0–2 activities) | 136 (28.9) | 80 (58.8) | 81 (84.4) | 71 (52.2) |
| Moderate (3–5 activities) | 87 (18.5) | 44 (50.6) | 47 (70.2) | 44 (50.6) |
| High (6–12 activities) | 118 (25.1) | 75 (64.1) | 72 (78.3) | 65 (55.1) |
| Very High (13–61 activities) | 130 (27.6) | 70 (53.9) | 66 (64.1) | 88 (67.7) |
| Dose–Mental Health (0–180 days) | | | | |
| Low (0 activities) | 110 (23.4) | 62 (56.4) | 61 (77.2) | 57 (51.8) |
| Moderate (1–2 activities) | 144 (30.6) | 77 (53.9) | 84 (79.3) | 77 (53.5) |
| High (3–6 activities) | 88 (18.7) | 50 (56.8) | 54 (75.0) | 51 (58.0) |
| Very High (7–43 activities) | 129 (27.4) | 80 (62.0) | 67 (66.3) | 83 (64.3) |
| Dose–Housing (0–180 days) | | | | *** |
| Low (0–2 activities) | 106 (22.5) | 60 (57.1) | 58 (77.3) | 49 (46.2) |
| Moderate (3–6 activities) | 122 (25.9) | 72 (59.0) | 71 (77.2) | 61 (50.0) |
| High (7–12 activities) | 94 (20.0) | 55 (58.5) | 55 (71.4) | 63 (67.0) |
| Very High (13–81 activities) | 149 (31.6) | 82 (55.0) | 82 (71.9) | 95 (63.8) |
| Dose–Social Services (0–180 days) | | | * | ** |
| Low (0 activities) | 142 (30.2) | 79 (55.6) | 76 (78.4) | 66 (46.5) |
| Moderate (1 activity) | 68 (14.4) | 36 (52.9) | 38 (80.9) | 39 (57.4) |
| High (2–5 activities) | 140 (29.7) | 80 (57.6) | 89 (76.7) | 92 (65.7) |
| Very High (6–33 activities) | 121 (25.7) | 74 (61.2) | 63 (64.3) | 71 (58.7) |
| Dose–Educational & Emotional (0–180 days) | | | ** | |
| Low (0–1 activities) | 111 (23.6) | 66 (60.0) | 67 (83.8) | 53 (47.8) |
| Moderate (2–5 activities) | 129 (27.4) | 69 (53.5) | 67 (71.3) | 75 (58.1) |
| High (6–13 activities) | 115 (24.4) | 69 (60.0) | 74 (78.7) | 69 (60.0) |
| Very High (14–80 activities) | 116 (24.6) | 65 (56.0) | 58 (64.4) | 71 (61.2) |
| Dose–Employment & practical support (0–180 days) | | | | |
| Low (0 activities) | 216 (45.9) | 131 (60.9) | 123 (77.4) | 118 (54.6) |
| Moderate (1–2 activities) | 113 (24.0) | 58 (51.3) | 66 (74.2) | 63 (55.8) |
| High (3–41 activities) | 142 (30.2) | 80 (56.3) | 77 (70.0) | 87 (61.3) |

\*p<0.15

\*\*p<0.05

\*\*\*p<0.01.

Total column results are displayed as n (column percent) for categorical.

Remaining column results are displayed as n (row percent) for categorical.

significantly associated with retention in care at 12 months, but there was no significant effect on stable housing or viral suppression. The type of navigation activity varied by housing stability and health outcomes. There was no significant effect of type of navigation activity on achieving stable housing at 6 months. There was some suggestive evidence that participants who received intensive employment and practical support (e.g., obtaining food, clothing, or cell phones) versus moderate employment and practical support had greater housing stability (approximately 56% versus 51%).

There was a significant association between the intensity of health care and education/emotional support activities with viral suppression. Those who received low (84.4%) to high (78.3%) health care support from navigators were virally suppressed compared to participants who received very high intensity of health care activities (64.1%). Similarly, participants who received low (83.8%) or high (78.7%) of education/emotional support were virally suppressed compared to those with very high intensity of education/emotional support (64.4%). Transportation and other social services also were significantly associated with viral suppression for those who received lower intensity (78.4%) and high intensity (76.7%) of services compared to those with very high doses (64.3%).

Finally, higher intensity of health care, housing and social service/transportation activities were significantly associated with retention in care at 12 months. Participants who received higher intensity of health supports (67% vs 52%), housing supports (64% vs 46%) and social service/transportation (59% vs 46%) were retained in care. In summary, the higher intensity of navigation activities in general led to greater retention in care. However, there were no significant effects of navigation on housing stability at 6 months and mixed effects on viral suppression depending on the intensity and type of services provided by the navigator (Table 2).

The final path model (Fig 2) depicts the strength of direct relationships between social determinants of health, housing stability, and subsequent retention in HIV care and viral suppression for all associations with significance level less than 0.15. In the final path analysis, our sample included 471participants with complete data. Table 3 provides all results of the direct relationships of the final path model. We found the path to stable housing at 6 months had a significantly increased likelihood for youth under the age of 30 years (AOR = 2.13 [95% CI 1.09, 4.18]), being virally suppressed at baseline (AOR = 1.56 [95% CI 0.95, 2.55]), longer time living with HIV (AOR = 1.27 [95% CI 1.01, 1.59]), and improved self-efficacy for obtaining help by 6 months (AOR = 1.36 [95% CI 1.06, 1.74]). Those with food insecurity (AOR = 0.57 [95% CI 0.35, 0.91]), moderate to higher risk of opiate use (AOR = 0.59 [95% CI 0.34, 1.03]), being transgender versus cisgender male (AOR = 0.34 [95% CI 0.15, 0.79]), and longer time being homeless (AOR = 0.78 [95% CI 0.66, 0.93]) were significantly less likely to be stably housed at 6 months.

Being stably housed at 6 months did not lead to a significantly greater likelihood of being retention in care at 12 months, but it did result in a greater likelihood of viral suppression at 12 months (AOR = 2.43 [95% CI 1.42, 4.16]). Viral suppression at baseline (AOR = 4.65 [95% CI 2.13, 10.16]) and being at moderate to high risk for opiate use (AOR = 1.98 [95% CI 1.53, 2.57]) were also associated with viral suppression at 12 months. Finally, a higher number of unmet needs at baseline (AOR = 0.65 [95% CI 0.55, 0.77]) and higher doses of patient navigation intervention: high versus low (AOR = 0.50 [95% CI 0.24, 1.03] and very high versus low (AOR = 0.55 [95% CI 0.36, 0.87]), were associated with a lower likelihood of viral suppression at 12 months (See Fig 2, Table 3).

Higher intensity of the patient navigation intervention was associated with an increased likelihood of retention in care at 12 months: high versus low dose (AOR = 2.36 [95% CI 1.11, 5.02]); very high versus low dose (AOR = 1.57 [95% CI 0.99, 2.48]). Retention in care was also associated with having greater social support (AOR = 1.23 [95% CI 1.10, 1.37], an increase in

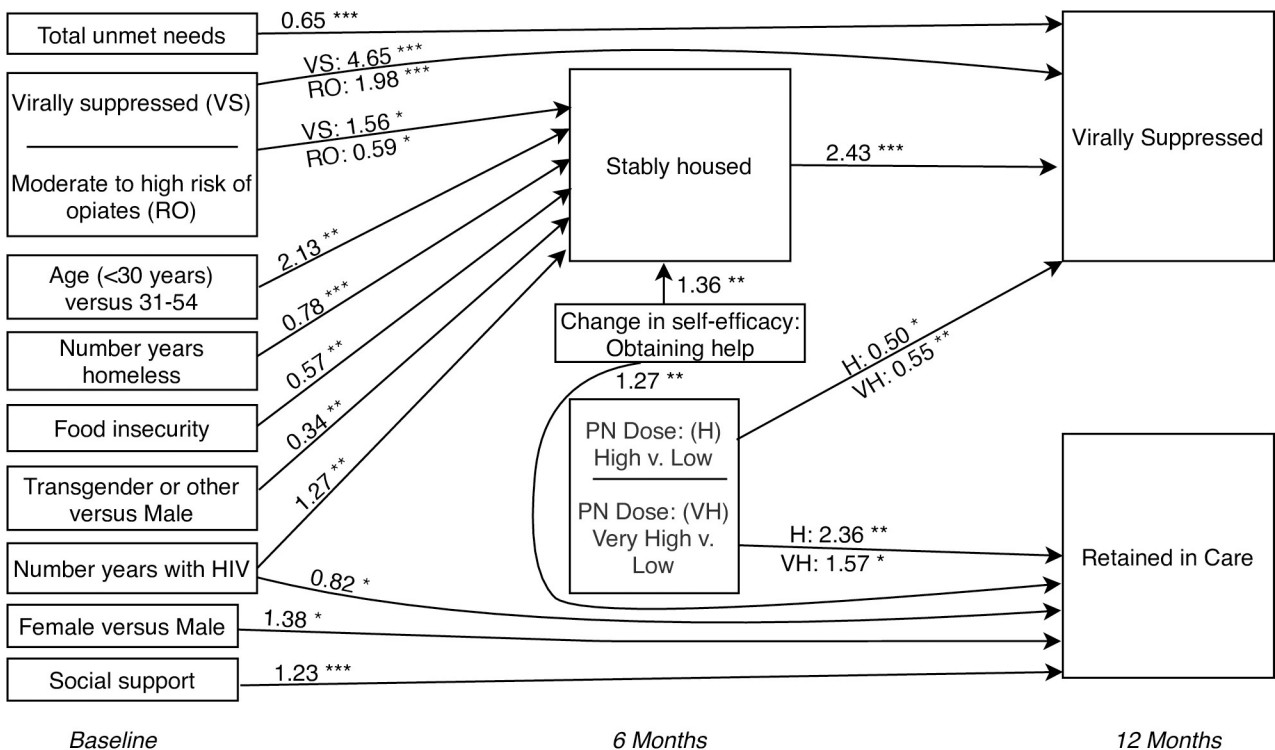

Fig 2. Path model for participant characteristics and navigation activity level on housing stability and HIV health outcomes.

self-efficacy score for obtaining help by 6 months (AOR = 1.27 [95% CI 1.05, 1.52]), and for cisgender female versus cisgender male (AOR = 1.38, [95% CI 0.98,1.93]). In addition, longer time living with HIV was associated with a lower likelihood of retention in care (AOR = 0.82 [95% CI 0.67, 1.01]). We found no direct effect of intensity of navigation activities on housing stability at 6 months and interestingly higher doses of navigation activities had a lower odds of viral suppression compared to those who received few contacts with navigators (Fig 2, Table 3).

## Discussion

This study examined factors associated with the pathways to housing stability and subsequent retention in care and viral suppression for people with HIV experiencing homelessness who participated in the HRSA/SPNS Homeless Initiative. We hypothesized that a higher dose of patient navigation activities would lead to increased likelihood of housing stability at 6 months, and thus, result in improved retention in care and viral suppression at 12 months post follow-up. While we found that navigation activities directly increased retention in care, there was no direct effect on stable housing. Similar to other studies, we found a direct effect of obtaining housing stability on viral suppression, yet, surprisingly, those with higher doses of patient navigation led to lower odds of viral suppression.

**Table 3. Path model logistic regression results for participant characteristics and navigation activity level with housing stability and HIV health outcomes, HRSA/SPNS Building a Medical Home for Multiply diagnosed HIV homeless populations 9 sites in the US, 2013–2017 (n = 471).**

| | AOR (95% CI) | p-value |
|---|---|---|
| **Stably housed at 6M (n = 269)** | | |
| Navigation Activity level—overall | | |
| Moderate vs low | 1.27 (0.70,2.28) | 0.428 |
| High vs low | 1.31 (0.61,2.82) | 0.484 |
| Very high vs low | 1.43 (0.88,2.35) | 0.152 |
| Food insecurity | 0.57 (0.35,0.91) | 0.019 |
| Age | | |
| 30 years or younger versus 31–54 years | 2.13 (1.09,4.18) | 0.027 |
| 55 years or older versus 31–54 years | 0.74 (0.38,1.42) | 0.364 |
| Number of years homeless | 0.78 (0.66,0.93) | 0.008 |
| Number of years with HIV | 1.27 (1.01,1.59) | 0.046 |
| Social support score | 1.19 (0.93,1.53) | 0.164 |
| Virally suppressed at baseline | 1.56 (0.95,2.55) | 0.077 |
| Change in self-efficacy score: Obtaining help | 1.36 (1.06,1.74) | 0.014 |
| Gender | | |
| Cisgender female versus cisgender male | 1.37 (0.79,2.37) | 0.266 |
| Transgender people or other identified versus cisgender male | 0.34 (0.15,0.79) | 0.012 |
| Number of unmet needs | 0.92 (0.80,1.06) | 0.223 |
| Number of barriers to care | 0.83 (0.66,1.04) | 0.107 |
| Moderate/severe risk for substance use: opiates | 0.59 (0.34,1.03) | 0.063 |
| **Retention in HIV Primary care at 12M (n = 268)** | | |
| Stably housed at 6 months | 1.00 (0.68,1.47) | 0.998 |
| Navigation Activity level—overall | | |
| Moderate vs low | 1.31 (0.51,3.41) | 0.575 |
| High vs low | 2.36 (1.11,5.02) | 0.026 |
| Very high vs low | 1.57 (0.99,2.48) | 0.054 |
| Food insecurity | 1.02 (0.66,1.58) | 0.913 |
| Ages | | |
| 30 years or younger versus 31–54 years | 0.96 (0.64,1.46) | 0.857 |
| 55 years or older versus 31–54 years | 1.13 (0.65,1.97) | 0.668 |
| Number of years homeless | 0.96 (0.82,1.13) | 0.610 |
| Number of years with HIV | 0.82 (0.67,1.01) | 0.064 |
| Social support score | 1.23 (1.10,1.37) | <0.001 |
| Virally suppressed at baseline | 1.15 (0.80,1.64) | 0.457 |
| Change in self-efficacy score: Obtaining help | 1.27 (1.05,1.52) | 0.011 |
| Gender | | |
| Cisgender female versus cisgender male | 1.38 (0.98,1.93) | 0.066 |
| Transgender people or other identified versus cisgender male | 0.79 (0.38,1.62) | 0.516 |
| Number of unmet needs | 0.90 (0.70,1.16) | 0.421 |
| Number of barriers to care | 1.12 (0.92,1.37) | 0.277 |
| Moderate/severe risk for substance use: opiates | 0.96 (0.53,1.74) | 0.890 |
| **Virally Suppressed at 12M (n = 266)** | | |
| Stably housed at 6 months | 2.43 (1.42,4.16) | 0.001 |
| Navigation Activity level—overall | | |
| Moderate vs low | 0.99 (0.62,1.58) | 0.955 |
| High vs low | 0.50 (0.24,1.03) | 0.061 |

(*Continued*)

**Table 3.** (Continued)

|  | AOR (95% CI) | p-value |
|---|---|---|
| Very high vs low | 0.55 (0.36,0.87) | 0.010 |
| Food insecurity | 0.58 (0.24,1.45) | 0.245 |
| Ages |  |  |
| 30 years or younger versus 31–54 years | 0.90 (0.45,1.81) | 0.760 |
| 55 years or older versus 31–54 years | 1.51 (0.75,3.02) | 0.247 |
| Number of years homeless | 1.02 (0.76,1.39) | 0.880 |
| Number of years with HIV | 0.90 (0.76,1.07) | 0.241 |
| Social support score | 0.81 (0.59,1.11) | 0.194 |
| Virally suppressed at baseline | 4.65 (2.13,10.16) | <0.001 |
| Change in self-efficacy score: Obtaining help | 0.99 (0.71,1.38) | 0.939 |
| Gender |  |  |
| Cisgender female versus cisgender male | 1.01 (0.61,1.68) | 0.955 |
| Transgender people or other identified versus cisgender male | 1.17 (0.18,7.69) | 0.874 |
| Number of unmet needs | 0.65 (0.55,0.77) | <0.001 |
| Number of barriers to care | 1.00 (0.68,1.49) | 0.991 |
| Moderate/severe risk for substance use: opiates | 1.98 (1.53,2.57) | <0.001 |

Adjusted odds ratio (AOR) and 95% confidence intervals (CI) for continuous variables are per standard deviation.

To our knowledge, this is the first study that examines the specific pathways for mechanisms of patient navigation programs to improve housing stability and HIV health outcomes among people with HIV who are unstably housed or homeless. Our results show that the intensity of navigation programs have a positive, direct effect on engaging and retaining this vulnerable population in medical care over time. Specific support for health care, housing, and transportation can lead to better outcomes. This finding highlights the critical role patient navigators can play as part of the care team in reducing barriers to care and keeping people with HIV experiencing homelessness connected to the health care system for continuity of care. Our findings are similar to those of other studies that found that patient navigation has a significant effect on engagement in care for vulnerable people with HIV at risk for homelessness, such as those leaving incarceration and those who have been out of care for more than six months [22–25].

Our participants included people with HIV with multiple co-morbidities, such as mental health and substance use disorders. In this sample population, we found initial suggestive evidence that reducing the unmet need for substance use treatment was associated with better housing stability (Table 1), however, there was no association with retention in care nor viral suppression in bivariate analyses. We, therefore, excluded it from the path model due to our selection criteria. Given these results, it is likely that structural effects, due to limited accessibility to substance use treatment and mental health providers, was a perpetual barrier that limited the effectiveness of navigation activities on viral suppression. Other studies have found an indirect effect of mental health coordination on reducing days to enter treatment for addiction services, specifically, as a critical factor for the pathway to addiction services, especially for racial ethnic minorities [26]. Further research is needed to assess how access to substance use and mental health treatment mediate housing stability and subsequent health outcomes.

The findings that the intensity of navigation activities did not have a significant direct effect on housing stability and resulted in a lower odds of viral suppression, surprised us. Our findings suggest that given the high level of unmet multiple needs in this population, patient

navigation addressed those needs first before housing stability could be directly achieved. Our study found that navigators provided an intensive amount of time on a variety of activities in addition to housing assistance including education about HIV treatment, managing disclosure and coaching on living skills as well as supporting employment related services, skills development, and providing practical support such as help with transportation, phones, clothing, and food. Similar to addressing substance use and mental health treatment, the structural factors of housing affordability and availability could not be addressed solely via the intensive work by patient navigators working with individual participants in a time-limited intervention of 6–12 months.

Finally, while navigation programs alone are not the magic bullet for reaching viral suppression, they addressed the intermediate factors in the pathway to promoting housing stability and retention in care, specifically for recently diagnosed people with HIV. Our population had multiple unmet needs in addition to stable housing including food, obtaining medications, substance use, and mental health treatment. Patient navigators may have spent more time addressing these priority unmet needs, and therefore had less time to devote to education and counseling on adherence to treatment. We found significant mixed effects of dose of education activities on viral suppression: those with very high doses (14–80 contacts) of education were less likely to be virally suppressed compared to those who received low to high doses (1–13 contacts) (Table 2). We could not tease out the specific educational topics or activities that made an impact or examine how sessions were delivered across sites. Our model also did not explore other factors that patient navigation activities could affect such as relationships with health care providers, stigma, and personal beliefs and attitudes towards addressing HIV that may impact reaching viral suppression. Further research on the effects of patient navigation models on these factors is warranted.

However, our findings signal other important elements for navigation programs that affect housing stability and viral suppression in this population. HRSA SPNS participants with moderate or high risk for opiate use were more likely to achieve viral suppression regardless of the person's ability to be stably housed. The HRSA SPNS intervention used mobile, interdisciplinary teams to address housing, behavioral health and medical care to help ensure people with HIV who were homeless had access to HIV medications and treatment and achieve viral suppression. These interdisciplinary teams of clinicians and navigators could be a key step in supporting a person's readiness to enter longer-term treatment at a health care facility or to become stably housed [27].

## Limitations

Our path analysis was dependent on a convenience sample of people with HIV experiencing homelessness. While we had fewer than 10% of missing data on outcome variables, we cannot rule out the possibility of selection bias on the outcomes for viral suppression and housing stability due to the nonrandomized, prospective study design. To address this bias, we conducted an attrition analysis of the characteristics of participants in this path analysis (n = 471) to those study participants lost to follow up by 12 months (n = 438). We found no differences on sociodemographics (age, race/ethnicity, gender and education), However this study sample were less likely to be chronically homeless (72% vs 78%), more likely to obtain food (67% vs 23%) and reported fewer unmet social and medical needs (3.4 vs 3.7), lower proportion of high risk for cocaine use (47% vs 57%) and slightly higher mental health functioning (35.8 vs 33. 8) [S1 Table].

Second, we only included those with permanent stable housing at 6 months as our mediator and did not consider other types of temporary or transitional housing models or perceptions

of housing security. Studies have found that other dimensions are important to include in measuring housing stability such as perception of security, safety, and quality of the housing [28]. It was beyond the scope of the study to capture how the HRSA SPNS intervention affected these dimensions of housing stability, which might subsequently impact retention in care or viral suppression.

Our measurement of patient navigation dose was driven by the data and we chose to divide intensity of contact into quartiles of "low", "moderate", "high" and "very high". We did not explore participant characteristics that may be related to each level of intensity, which could account for the direct effect of the intervention on retention in care and viral suppression. Further research in the area is needed to understand if specific characteristics are related to the intervention dose.

Finally, we collected data on housing status and other enabling and risk factors (food security, opiate risk, self-efficacy) via self-report in interviews and intervention dose at 6 months to see the impact on clinical outcomes at 12 months. Because some of these factors were measured at the same time, it could account for some of the null effects on stable housing as some of these other variables are serving as mediator effects. Further studies should explore the temporality of these factors on housing status and subsequent clinical outcomes.

Despite the above limitations, these findings highlight the potential impact for the scope and practice of the navigation activities and the need for greater resources to support system coordination in working with people with HIV who experience homelessness. First, prioritizing the needs and developing a person-centered care plan to address and monitor the progress with addressing multiple needs is important. As part of the medical home, the HRSA/SPNS interventions worked with an interdisciplinary team to create a coordinated plan for addressing medical and social needs including housing. Our path analysis revealed specific populations who benefitted from the intervention, such as youth under the age of 30, women, recently diagnosed, and persons with high risk for opiate use. Other populations, such as transgender people, were less likely to obtain stable housing. While our transgender population was a small percentage, our findings highlight a need to provide culturally appropriate training for staff in better response to the needs of transgender people and also serve as an advocate with housing providers and landlords to reduce possible stigma and discrimination.

Second, the model revealed key indicators associated with stable housing, retention in care, and viral suppression, such as increased social support and improved self-efficacy with obtaining help. Providing training and supervision to patient navigation staff on how to discuss, motivate, and support clients and provide services that promote client self-efficacy in this area could lead to better housing and HIV care outcomes.

Finally, there is a greater need for system coordination to address the accessibility and affordability of housing, access to employment, substance use, and mental health treatment. It was beyond the scope of this study to examine or measure the level of system coordination on housing and HIV health outcomes, however, this would be an important next step in understanding how barriers to care are reduced and housing stability may be improved.

## Conclusions

In summary, we found in the sample of people with HIV experiencing homelessness participating in a navigation intervention, housing stability was a significant direct pathway for viral suppression. Population groups varied in the pathway to housing stability and health outcomes. Youth, and people with HIV who were more recently homeless were more likely to achieve stable housing. Women and people who were newly diagnosed with HIV had a greater odds of being retained in care, and people with HIV with a moderate to high risk of opiate use

were more like to achieve viral suppression. Transgender persons and those experiencing high food insecurity were less likely to reach housing stability and subsequent improvements in HIV health outcomes. Navigation activities did not have a direct effect on the pathway to housing stability, but they were directly related to retention in care. The results identify key populations and factors to target resources for social interventions and policies to achieve improved housing stability and HIV health outcomes.

## Supporting information

**S1 Table. Study sample attrition analysis, HRSA/SPNS Building a Medical Home for Multiply-Diagnosed HIV-positive Homeless Populations initiative from 2013–2017.** (DOCX)

## Acknowledgments

The authors wish to thank all the intervention and evaluation staff, care coordinators, patient navigators and clients who participated in the HRSA/SPNS Building a Medical Home for multiply diagnosed HIV-positive homeless populations Initiative and the following study group members:

Serena Rajabiun, PhD, Sara S. Bachman PhD, Howard Cabral, PhD, Jane Fox, MPH, Emily K. Quinn, MA, Mariana Sarango, MPH, Carmen Avalos, MD, Alexander de Groot, MPH, Karen Fortu, MPH, Kerrin Gallagher, MPH, Boston University; Barbara Cocci, MSW, Carole Hohl, PA-C, Sandy Sheble-Hall, BSN, Boston Health care for the Homeless Program.

Manisha H. Maskay, PhD, Nicole Chisolm, MPH, Ben Calloway, LMSW, Prism Health North Texas.

Bahby Banks, PhD, MPH, Ayodele Gomih PhD, MSPH, Lisa McKeithan, MS,CRC Commwell Health.

Verna Gant, MBA, Family Health Centers of San Diego; Amy Pan PhD, San Diego State University.

Tom Giordano, MD, MPH, Jessica Davila, PhD, Baylor College of Medicine; Nancy Miertschin, MPH, Siavash Pasalar, PhD, Harris Health System.

Jo Ann Davich, BA, Multnomah County Health Department.

Angelica Palmeros, MSW, Matt Feaster, MPH, Pasadena Public Health Department.

Deborah Borne, MD, MSW, Janell Tryon, MPH, San Francisco Department of Public Health; Kate Franza, Asian & Pacific Islander Wellness Center.

Mobeen H. Rathore, MD, Kendall Guthrie, M Div, University of Florida Center for HIV/ AIDS, Research, Education & Service (UF CARES).

Frederick Altice, MD, MA, Ruthanne Marcus, PhD, Mary L. Powell, DNP, PMHNP-BC, Yale University AIDS Program; Silvia Moscariello, MBA, Liberty Community Services.

Melinda Tinsley, MA, Pamela Belton, Thelma Iheanyichukwu, MHA,Chau Nguyen, MPH, Corliss Heath, PhD, MPH, Health Resources and Services Administration, HIV/AIDS Bureau.

## Author Contributions

**Conceptualization:** Serena Rajabiun, Howard J. Cabral.

**Data curation:** Emily K. Quinn, Deborah Borne, Manisha H. Maskay, Thomas P. Giordano.

**Formal analysis:** Serena Rajabiun, Kendra Davis-Plourde, Emily K. Quinn, Howard J. Cabral.

**Funding acquisition:** Serena Rajabiun, Melinda Tinsley.

**Investigation:** Serena Rajabiun, Deborah Borne, Manisha H. Maskay, Thomas P. Giordano, Howard J. Cabral.

**Methodology:** Serena Rajabiun, Kendra Davis-Plourde, Melinda Tinsley, Howard J. Cabral.

**Project administration:** Serena Rajabiun, Melinda Tinsley.

**Supervision:** Serena Rajabiun.

**Writing – original draft:** Serena Rajabiun, Kendra Davis-Plourde, Howard J. Cabral.

**Writing – review & editing:** Melinda Tinsley, Emily K. Quinn, Deborah Borne, Manisha H. Maskay, Thomas P. Giordano.

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
