## [Decision Letter · Decision Letter 0]

4 Feb 2020

PONE-D-19-29536

Pathways to housing stability and viral suppression for people living with HIV/AIDS: Findings from the Building a Medical Home for Multiply Diagnosed HIV positive Homeless Populations Initiative

PLOS ONE

Dear Dr. Rajabiun,

Thank you for submitting your manuscript to PLOS ONE. After careful consideration, we feel that it has merit but does not fully meet PLOS ONE’s publication criteria as it currently stands. Therefore, we invite you to submit a revised version of the manuscript that addresses the points raised during the review process.

As you can see from the reports below, three reviewers provided positive comments regarding the scientific rationale and relevance of your study, but they also raised several concerns that need to be fully addressed in order for the manuscript to be further considered for publication. 

We would appreciate receiving your revised manuscript by Mar 19 2020 11:59PM. To enhance the reproducibility of your results, we recommend that if applicable you deposit your laboratory protocols in protocols.io, where a protocol can be assigned its own identifier (DOI) such that it can be cited independently in the future. For instructions see: http://journals.plos.org/plosone/s/submission-guidelines#loc-laboratory-protocols

We look forward to receiving your revised manuscript.

Kind regards,

Dario Ummarino, Ph.D.

Academic Editor

PLOS ONE

Journal Requirements:

2. Thank you for stating in the manuscript: "All study procedures were approved by local Institutional Review Boards at the nine participating study sites and the multisite evaluation center at Boston University Medical Campus. The Office of Human Research Protection at the Department of Health and Human Services granted a certificate of confidentiality for the study."

Reviewers' comments:

Reviewer's Responses to Questions

**Comments to the Author**

1. Is the manuscript technically sound, and do the data support the conclusions?

Reviewer #1: No

Reviewer #2: Partly

Reviewer #3: Partly

2. Has the statistical analysis been performed appropriately and rigorously? 

Reviewer #1: Yes

Reviewer #2: I Don't Know

Reviewer #3: Yes

3. Have the authors made all data underlying the findings in their manuscript fully available?

Reviewer #1: Yes

Reviewer #2: Yes

Reviewer #3: Yes

4. Is the manuscript presented in an intelligible fashion and written in standard English?

Reviewer #1: Yes

Reviewer #2: Yes

Reviewer #3: Yes

5. Review Comments to the Author

Reviewer #1: The submitted manuscript describes an analysis of the effect of patient navigation on people living with HIV (PLWH) who are homeless and are diagnosed with co-occurring substance abuse or mental health disorders. Participants were part of one of the Health Resources and Services Administration’s Special Projects of National Significance: “Building a Medical Home for Multiply-Diagnosed HIV-positive Homeless Populations Initiative 2012-2017 (HRSA SPNS Homeless Initiative), which found that navigation models may be an effective intervention to support people who experience homelessness achieve more stable housing, improve retention in care, and reach viral suppression. However, little is known about the mechanisms by which patient navigation interventions work to help people achieve stable housing and subsequently improve health outcomes. This study was set out to examine participant and program factors for achieving stable housing at 6 months and how these factors influenced retention in care and viral suppression at 12 months. In total 700 unstably housed PLWH enrolled in the patient navigation intervention across nine sites in the United Stated from 2013-2017. A path analysis model with housing stability at 6 months as the mediator was used to examine the direct and indirect effects of participant’s socio-demographics and risk factors and patient navigation on viral suppression and retention in care at 12 months. Significant effects of patient navigation intensity on retention in care outcomes at 12 months are reported, however patient navigation intensity was not associated with housing stability at 6 months and was negatively associated with viral load suppression at 12 months. While the manuscript provides critical data for our understanding of the underlying mechanisms impacting housing stability and HIV outcomes among this vulnerable population, several inconsistencies across the paper need to be addressed before publication can be considered.

Specific comments:

• Instead of “people with HIV” would use more established and accepted acronym “PLWH” (people living with HIV)

• Figure 1 and 2 – legends are missing

• Figure 1 – need to elaborate on enabling factors (predictors) – “socio-demographic” does not seem to capture all

• A brief description of the 9 study sites would be helpful

• Results are inconsistent across abstract, tables, text – for example abstract states that being male was associated with housing stability at 6 months, however tables indicate being female was associated with housing stability. This inconsistency also impacts the conclusions of this manuscript.

• Regarding depression instrument used – not clear which depression screening instrument was used – short form or long form!?

• One of the more surprising findings is not discussed sufficiently, namely the negative association of patient navigation intensity and VL suppression – maybe patients struggling with VL suppression were given more resources/attention!? However, this also warrants a more detailed description on how patient navigation was provided in this population.

• References are very few, which is surprising given the body of literature by HL Cooper and O Galarraga.

Reviewer #2: Paper Ref. PONE-D-19-29536

Paper title: Pathways to housing stability and viral suppression for people living with HIV/AIDS: Findings from the Building a Medical Home for Multiply Diagnosed HIV positive Homeless Populations Initiative

Overview

This is an interesting study carried out in socially and economically excluded people such as people living with HIV and experiencing homelessness. The study tested the factors associated with housing stability at 6-month of follow-up and HIV viral suppression status and retention in care at 12 months follow up. The study also explored the mediating role of housing stability status at 6 month on the HIV viral suppression status and retention in care at 12 months follow up. The authors found that some individual-related factors were associated with housing stability at the 6th month of follow-up, which in turn have a positive effect on viral suppression at 12 months of follow-up. They also found that the intensity of the navigator-related activities has a negative impact on viral suppression at the 12th month of follow-up.

Overall, I find that this study provides an important insight into the role of archiving housing stability on health-related outcomes such as HIV viral suppression. These findings also support the need to enhance and provide access to housing and health and social support services for people experiencing homelessness with or without comorbid mental and physical health-related problems. However, I have the below comments to the authors, which could improve the quality of the paper before it can be accepted for publication.

Abstract

1. Please define in short what a navigator model is.

2. Please state what those nine sites involved in the study were, as it will provide a better study contextual background for potential readers are were.

3. In the abstract, some dot marks are missing (i.e., lines 44 and 56).

4. Line 69, the avocation for policies, should not only be limited to clinical intervention and policies. It should also include social interventions and policy to address both the health and social unmet needs (lack of access to housing) of these population groups.

Introduction

1. Please elaborate more on the idea behind the statement “housing stability is an area that still needs progress.”

2. In lines 86 to 90, you summarized the already known effect of housing stability on appropriate HIV care retention and viral suppression in your study sample. Thus, you should state clearly how this new study differs from that one, especially as you also looked at the effect of housing stability on HIV care retention and viral suppression.

3. Line 99. It would be helpful to describe what a navigator model is.

4. The objectives of the study should state clearly the mediating effect of housing stability on health-related outcomes.

5. Relating to the follow-up period, (lines 102, 103) considering that the HRSA SPNS Homeless Initiative lasted from 2012 to 2017, why the analyses were only focused on the first year of follow-up, rather than over the 5 year follow-up period. These should be acknowledged and explained.

Methods

1. You stated that the HRSA SPNS was carried out in nine sites, could you please provide information on which were those nine study sites? This will help to give more background to the study and help the reader to interpret the findings based on the study context.

2. Line 116, define or characterize who the patient navigators were. It will help to understand the nature and dynamics of the intervention.

3. Please expand more on what specific housing-related services the navigator program gives to the participants (line 119). Does it include access to housing accommodation? Rent supplements?

4. Line 119, how many participants were initially enrolled?

5. Line 123. Please, expand more on the criteria used to define the unstably housed status.

6. “Unmet need for services” (line 128), is it referring to social or health services? Or both?

7. Inline 133, you stated that 700 participants were included. However, for the final path analysis, you only included 471 participants. Please state the actual number of participants included, and or whether you used different samples for some analyses. If you final sample was 471 participants, please provide information on potential clinical and demographic differences between these 471 and those excluded from the analyses.

8. Measures: As I said before. One of the main questions is: why did you only carry out the study analyses on the first-year follow-up period and not over the five-year follow-up period? As the housing stability status and intervention doses during the first six-month of follow-up could vary from those of the 6 to 12-month follow-up, impacting the outcome analyzed. Thus, it is important to explain why you did not also consider those measures for the 6 to 12 months period. As for some people experiencing homelessness, it is hard to achieve housing stability in a short period, even when access to housing is facilitated. Also, those people having higher intervention doses during the 6 to 12 months may have archived better health-related outcomes. I understand that this is a pathway analysis; however, there are statistical tools to perform longitudinal pathway analyses using time-varying mediators (See: Zheng et al. Longitudinal Mediation Analysis with Time-varying Mediators and Exposures, with Application to Survival Outcomes. J Causal Inference. 2017;5(2). pii: 20160006. VanderWeele et al. Mediation analysis with time varying exposures and mediators. J R Stat Soc Series B Stat Methodol. 2017; 79(3): 917–938)

9. In the measure subsection, the definition and operationalization of all co-variates considered in the analyses should be described.

10. Lines 155-158. It would be more informative, to describe some examples of specific mental health, housing, social, transportation-related activities that are included in these six program domains.

11. Relating to intervention dose, did you explore its effect by grouping those related to health care supportive services vs non-health related supportive services? As by considering all into one variable may hide the potential positive effect of health-related intervention activities on the HIV viral suppression outcome. Also, the intervention dose for the health-, housing-, social service-, education-, and employment-related areas may have a distinct effect on the outcome analyzed.

12. What about the access to pharmacological viral-related treatment. Does this population have access to the existing effective HIV treatments? Do you have any information about it? If not, this is an essential limitation of the Intervention program that should be acknowledged.

Results

1. In table 1. Those all covariates you presented should be described in the Methods section (as I previously commented). In addition, you could explore the dose of health-related services (health care, mental health care) and the dose of non-health related services (housing, social services, education and employment) on the analyzed health outcomes outcome.

2. Line 220. As said before, the actual number of participants included in the study should be clearly stated. As if for some analyses, you used one sample, and for others, you used a smaller sample, somehow, you are analysis two population groups. Considering present a consistent sample (N=471) and describe any potential differences with the sample of participants who were not included in the final analyses.

Discussion

1. 281. Please expand what a patient navigator actor(s) is/are. His may help the understanding of the potential reader who is providing these kinds of services or activities. Expand the background of these programs.

2. Line 921. Please elaborate on what unmet needs you are referring to (social, mental, physical health, emotional, or all of these).

3. Line 305. Could you comment on whether the navigation-housing related activities also facilitate immediate access to housing? This can be one of the explanations of why the dose of activities was not related to housing stability and viral suppression. For example, the Housing First programs have shown to be effective in helping people to reach housing stability in short and long-term periods. People experiencing homelessness have complex and multidimensional health, social, and emotional unmet needs, which require strong evidence-based interventions and permanent support to observe changes in different dimensions. However, the immediate access to housing without mental, substance use or other social requirements should be provided and should be advocated, rather than wait until those complex needs are addressed before a person could access to stable housing. Thus, could you restate what you are suggesting on line 305/306?

4. Line 312. Could you extend more of which additional supportive services will increase the housing stability and better HIV-related outcomes in these populations? Example, Housing First interventions or other evidence-based initiatives.

5. Line 319. Are the mixed effects of education activities on viral suppression presented somewhere on the manuscript? If not, it would be great to include them in the results section or supplemental information.

6. Limitation: Did the study participants have access to HIV pharmacological HIV treatment as part of the navigator intervention activities? If not, it is a significant limitation as it can influence the lack of positive effect of the intervention dose on HIV viral suppression.

End of comments

Reviewer #3: This paper reports findings from a SPNS project on patient navigation at medical homes around the US. The authors conducted a path analysis to assess whether housing stability at six months mediated the relationship between baseline individual demographic and clinical characteristics and retention in care or viral suppression at 12 months. This analysis addresses important issues about whether navigation services and housing stability influence HIV medical outcomes and for whom they work. More literature on this topic would be valuable and of broad interest. This paper has several issues, some of which are serious, and all of which are amenable to changes:

1. The abstract is missing at least three periods, e.g., “…and reach viral suppression However, there is…”

2. Navigation and models of it are not described in the Intro and should be.

3. Line 93, “statistically significant”: Please clarify what was tested that was significant.

4. Lines 115-119: So intervention sites were primary care providers? Are the three mentioned characteristics what also define a medical home? A general working definition of “medical home” is warranted.

5. Lines 134-135: Although it may be in other sources, please in this paper clarify how retention and suppression at 12 months were defined (both the time range for the care/VL and the data source), particularly because you have more people suppressed than retained in care, which is a bit counter to the standard continuum of care.

6. Table 1: In column headers, please add what the percents are. For example, the “Total” column clearly has column percents – please expand headers to make the other percents easier to interpret.

7. Table 1: Please add a total row.

8. Table 1: Can you comment in the results about how so many more people are virally suppressed than retained in care at 12 months? For example, for males, 74% were suppressed and only 53% retained in care. Seems unusual – or maybe your column headers were switched?

9. Table 1: I suggest using the same groupings but different labels for your gender variable. For example, “Cisgender man,” “Cisgender woman,” and “Transgender or another gender identity.” This avoids male/female terms for gender and the suggestion that transgender persons don’t identify with a binary gender.

10. Lines 298-334: Numerous points in the discussion seem to reveal your bias that patient navigation must be effective. For example, “…was a perpetual barrier that limited the effectiveness of navigation activities” (line 298-299), “patient navigation is required to address those needs first before housing stability can be directly achieved” (lines 304-306), “other important elements for patient navigation that affect housing stability and viral suppression” (lines 327-238), and “mobile interdisciplinary teams of clinicians and patient navigators are a key step” (line 332-333). More data on patient navigation would be very useful in the literature, so your focus is appreciated. But it would be more convincing if you wrote as though you didn’t have this bias and patient navigation could, possibly, be ineffective at the things you failed to prove in your analysis. Most importantly, provision of navigation activities isn’t randomized, so the people struggling the most (with a range of things, including those that make achieving housing stability and your medical outcomes difficult) may be offered the most help. Your second Limitations paragraph (lines 346-351) almost addresses this, but not quite, and should. It also seems like there should be a citation out there that you could call on about this apparent paradox, that dose of services and good outcomes are not necessarily correlated.

11. Line 332: “Across” instead of “crossed”?

12. Lines 328-329: Unclear how this sentence relates to the rest of the paragraph – what do you make of this finding?

13. Line 331: First time mentioning “peer” – introduce and define in Intro.

14. Lines 332-333: Would be helpful to mention this care/treatment delivery mode in Intro or Methods. Unclear which aspect(s) of this care/treatment delivery mode influences outcomes. The team was mobile, and it was also a clinician and navigator. Was the peer / navigation aspect of the team truly impactful? From this analysis, it seems that we can’t say.

15. Line 358: “Of” instead of “for”?

16. Lines 384-385: I don’t think your written Results section supports this statement, e.g., for transgender or food-insecure persons. Would be helpful to clarify which “benefit” is being discussed and also to ensure that any results important enough to highlight in Conclusions are written in the Results section.

6. PLOS authors have the option to publish the peer review history of their article (what does this mean?). If published, this will include your full peer review and any attached files.

Reviewer #1: No

Reviewer #2: No

Reviewer #3: No

---

## [Author Response · Author response to Decision Letter 0]

4 May 2020

Response to Reviewer 1

• HRSA’s policy for written communications, including manuscripts requires that we use “people with HIV” and avoid using acronyms such as “PLWH”.

• We have added appropriate titles and legends to Figures 1 and 2. Figure 1 visually depicts our framework for the study. The title was somehow omitted when embedded in the manuscript and now has been added. In Figure 2, there is no legend associated with the path model. However, we revised the Figure to avoid acronyms such as “AOR” and changed to “adjusted odds ratio”.

• The abstract has been revised to ensure consistency with the tables and the manuscript. Table 1 shows the unadjusted bivariate analysis that select the criteria variables for the final path analysis. In unadjusted analyses, gender was a significant factor associated with being stably housed at 6 months and retention in care and therefore, included in the final analytic model. Male gender is the reference group and our findings indicate that transgender persons were significantly less likely to be stably housed at 6 months compared to males, while females were more likely to be retained in care at 12 months compared to males. The abstract has been revised to exclude the reporting of the unadjusted bivariate analyses and only include the findings from the final path model. 

• Depression screening: we have updated the methods section to include that we used the short form 10-item Centers for Epidemiologic Studies-Depression Scale.

• Findings related to patient navigation and viral suppression: Thank you for your comment about the surprising negative finding that persons who received higher doses of patient navigation were less likely to be virally suppressed in the final analysis. We have provided more detail in the methods and discussion section about how patient navigation was conducted based on the findings in Tables 1 and 2 and the final path analysis in Figure 2. We hypothesize that patients with whom patient navigators were providing a greater frequency of contact and variety of support services (health care, behavioral health support, other social service needs) had other priority needs. We separated our initial Table 1 into two tables: Table 1: Sociodemographic and health characteristics of the study sample and Table 2: Type and Intensity of Navigation Activities by Housing and Health Outcomes. Our findings in Table 2 show that navigators provided a significant amount of health care related services which included education and support with medications. One of the limitations of our study is that we did not collect data on whether patient navigators specifically observed patients taking medications which would more directly lead to viral suppression. It was also beyond the scope of this study to tease out if participants who received navigation support with obtaining medications had also received directly observed therapy compared to those who did not have better viral suppression. Our encounter form for navigators only documented “assist with obtaining medications”. This is an area for further study. 

• Thank you for the suggestion to review the studies published by HL Cooper and O Galarraga. We conducted a specific literature review to focus on people experiencing homelessness and HIV. We added the relevant studies from Galarraga and other recently published in 2018.

Reviewer 2

• For the abstract we added a definition of patient navigator and a description of the sites to give more context for the study (lines 29-30). In addition, we added a dot marks when appropriate. Thank you for these edits.

• In line 54 we revised the sentence to mention that interventions address the unmet social and health needs for this population and not only biomedical interventions. 

• Introduction: We revised the section to elaborate that while there is ample evidence that unstable housing affects HIV health outcomes, evidence is needed on the types of interventions and policies and the mechanisms of how they address housing instability and health outcomes. We also revised this section so it clearly addresses the gap in the research: examining how patient navigation can affect housing stability and viral suppression and if there is a direct pathway. 

o Added a description of a navigator model (lines 69-71).

o Added the objective of the study to clearly state that we are examining the mediating effect of housing stability on health-related outcomes (lines 86-89).

o The analyses for this study use outcomes based on HRSA core indicators of viral suppression at 12 months and retention in care at 12 months. Therefore, participants were enrolled in the study and followed for a minimum of 12 months and sites were required to collect data on participants at least to 12 months post enrollment. 

• Methods: We added more description about location and organizational settings for the nine sites. We had cited the original study which provides this background information to save on word count. We also added more description about the patient navigation model and the types of services, specifically housing services that were provided. Our encounter form collected information about housing search, accompaniment to housing support services and rental assistance, emergency housing services for motel stays; move in support. Our program intervention did not directly provide rental assistance but more navigators helped to connect participants to these programs including housing vouchers and other subsidy programs.

o Line 111: we revised to state that across the project 909 participants were enrolled and this study focuses on the subsample with 471 with complete housing and HIV health outcome data up to 12-month post intervention.

o Line 116: Our criteria used to define “unstably housing status” is based on HUD’s definition of homelessness and unstable housing. We have provided the citation and updated the manuscript to include that description. 

o Unmet need for services includes both social and other non-medical needs as well as medical needs (lines 168-171)

o We have updated our analysis to only focus on the sample with complete data n=471. We have also included a statement about the difference between this sample and those enrolled (n=438) (See line 456 -462) who did not have complete data on housing stability or health outcomes due to being lost to follow up at 6 or 12 months. We found the samples to be similar on sociodemographic variables (age, race/ethnicity, gender, education) but our final analytic sample had fewer social and medical unmet needs and barriers to care, and fewer chronically homeless persons compared to the overall samples with incomplete data (n=438). We have also included a table in the Supplementary Information section describing these differences.

o Statistical Analyses (lines 178-193): Thank you for this suggestion and citation. Our main objective of the study was to examine housing stability as a mediator on health outcomes. Therefore, we wanted to see if achieving housing stability and dose of patient navigation at 6 months led to better outcomes at 12 months. We used current standard software for path analysis, M-plus, that can incorporate time varying mediation. We realize that it is challenging to achieve housing stability in 6 months. We wanted, however, to examine intervention effects that occurred prior to our outcomes to better understand the potential mechanisms of action of the intervention. If our intervention and outcomes were occurring at the same time point (12 months) it would be more difficult for us to tease out the intervention mechanism. 

o Measures lines 135-196: we have included more information and description about all our measures. We had included a citation to a previous published study that has all the measures defined to save on word count. However, this section is now updated per the reviewer’s suggestion to include all the measures and appropriate citations.

o Navigation activities lines 143-158: We have included more description of the activities provided under each program domain. 

o Comment #11 on intervention dose: Our goal in this paper was to explore the global effects of patient navigation and dose on housing stability and health outcomes. It was beyond the scope of the paper to examine the types of activities. We wanted to keep the paper focus to the questions of overall dose. We are planning to do a more in-depth exploration for patient navigation activities in this population in the future. 

o Pharmacological viral treatment: We did not collect information on the specific pharmacologic treatments. This paper focuses on navigation and linkage to housing and HIV health outcomes: retention and viral suppression. 

• Discussion: We have revised the manuscript to give more understanding about patient navigation and who the patients are and the types of services provided. We have also included more description of the unmet needs, both non-medical and medical. 

o Line 309-319: We have revised this section and commented on whether navigation activities facilitate immediate access to housing. In table 3 our logistic regression shows that navigation dose at any level led to greater odds of housing stability, although there was no significant difference between those who received a lower dose versus those who received a higher dose. We more clearly explain that our navigators provided multiple services and it is difficult to tease out if one type of service made a difference on housing stability (see Table 1). In bivariate analyses, types of services seemed to more related to health outcomes viral suppression and health care support and access to housing and retention in care.

• Line 218-235: We have revised to include a description based on bivariate analyses that health care related services and educational and emotional support are more likely to impact viral suppression. There was no one type of service (Table 2) that was more related to housing stability, as it appears all activity was related although we did not find a significant association. This could be attributed to the short term of 6 months time frame that we investigated.

• Line 319: Yes, we included the mixed effects of educational support on viral suppression in the manuscript. Please see lines 226-228.

• Limitations: Yes, participants had access to pharmacological treatment. Although we did not collect data on their specific medications—this was a patient center medical home and the role of the navigator was to connect participants to HIV medical care (a prescribing provider) and housing

Reviewer 3:

• Comments #1 & 2: We have revised the abstract and edited for appropriate punctuation such as removing or adding extra periods per the Reviewers 2 & 3 suggestions.

• Comment #3: We have updated the introduction to provide more detailed description of Navigation models.

• Comment #3: Line 69-71, the test for statistical significance has been added

• Comment # 4: Lines 99-102: A general definition of the medical home intervention is included.

• Comment #5: Lines 135-138: The definition for retention in care and viral suppression are defined under measures in the methods section of the paper. The time for both measures is at 12 months post intervention and we have added a citation for both measures. The viral suppression measure is the standard measure used by HRSA. Retention in care has several definitions we have selected to use the one recommended in the paper by Mugavero. From our perspective this definition would seem important for continuity of care for people with HIV who are experiencing homelessness and a co-morbid condition such as mental health and/or substance use. 

• Comments # 6 & #7 on Table 1: We have revised the table to include row percentages per the reviewer’s suggestion. In addition, we have clearly labelled all Column headers to include N (%). 

• Comment # 8: Thank you for your question about our finding that there was a higher percentage of participants virally suppressed (75%) vs retained in care (for males 75% were virally suppressed and 55% were retained in care). We hypothesize this is attributable to the fact that some of our sites had medical home teams that were “mobile” with a physician, navigator, nurse and/or social worker. Some participants were seen by a prescribing health care provider in the community, were followed up with their antiretroviral medication consistently, and then connected with a four walls clinic such as community health center (with which many of these mobile teams were affiliated or had partnerships). Furthermore, we used a conservative definition of retention in care in which a person had to be seen at the clinic at least two appointment 90 days apart in a 12-month period. Some participants may have only been seen at 6 months and then could have received their ART prescription for 6 months. Unfortunately, we were not able to obtain from chart review the date of last prescription. We only had information if they had a recent prescription documented in their chart (yes/no).

• Comment # 9: We have updated the Gender variable in Table 1 to include the titles: “cis male” and “b cisgender female”.

• Comment #10: lines # 324-336: The manuscript has been revised to more neutral language and we removed the perceived bias towards patient navigation activities. In addition, we have expanded discussion in the second paragraph about the lack of randomization in the study. Unfortunately, we were not able to find a citation about the higher need patients needing the most help and struggling to achieve housing stability and viral suppression. 

• Comment #11: The sentence has been revised to say “across” instead of “crossed”. Thank you for finding this error.

• Comments #12-14: This section has been revised to remove any reference to peer navigation models so as not to speculate differences with this model versus other navigation models since it was not the scope of the paper. The scope of this paper was to examine dose of navigation activities and a future paper will examine the team composition. 

• Comment #15: The sentence has been changed from “for” to “of”. 

• Comment #16. The results section (lines 254-255) indicate the findings for transgender and food insecure persons. 

Thank you for the detailed review of our manuscript and the opportunity to make revisions. We look forward to the opportunity to publish our study in PLoS ONE.

---

## [Editor Report · Decision Letter 1]

11 Jun 2020

PONE-D-19-29536R1

Pathways to housing stability and viral suppression for people living with HIV/AIDS: Findings from the Building a Medical Home for Multiply Diagnosed HIV positive Homeless Populations Initiative

PLOS ONE

Dear Dr. Rajabiun,

Thank you for submitting your manuscript to PLOS ONE. After careful consideration, we feel that it has merit but does not fully meet PLOS ONE’s publication criteria as it currently stands. Therefore, we invite you to submit a revised version of the manuscript that addresses the points raised during the review process.

We look forward to receiving your revised manuscript.

Kind regards,

Laramie Smith, PhD

Academic Editor

PLOS ONE

Additional Editor Comments (if provided):

Dear Serena,

Many thanks for your detailed response to the reviewers comments. I believe this paper is positioned to make important contributions to the field.

From my read there is only one minor issue that could enhance the contributions of the publication in its current form. I believe R2 had asked for a legend in Figure 2, but that a bit more specificity may have helped in their request. I believe this figure presents the heart of this manuscript, and readers, like me may want to jump right to the figure to understand what the study found before they read all of the study methods. I would recommend that the figure legend be expanded on, in addition to specifying how results are presented. Specifically, it is not clear in the figure what H and VH are referring too. Figure 1 lets me know this is your patient navigation intervention dose, but if I'm just looking at Figure 2 I miss that this is (a) intervention dose, and (b) what the intervention dose is (i.e. patient navigation).

Could you, for example, label that box PN Dose: H: High v. Low -- VH: Very High v. Low, and then in the legend spell out that PN = Patient Navigation, PN Low dose: (give brief description of this dose), PN High dose: (give brief description of this dose), PN Very High dose: (give brief description of this dose)? That way a reader has all the information they need to interpret this figure in one spot. I think this minor revision will substantially help the interpretability of study findings.

---

## [Author Response · Author response to Decision Letter 1]

12 Jul 2020

We have updated Figure 2 to include a legend with a description of the patient navigation dose. We hope this addresses the editor's comments and makes the diagram more easily interpreted for the reader.

In response to an email dated 7/2/2020, a revised version of the manuscript has been uploaded with the file name "Manuscript 7-20" which is laid out in Portrait orientation not landscape with one-inch margins on all sides. Please advise if further changes are necessary.

---

## [Editor Report · Decision Letter 2]

2 Sep 2020

Pathways to housing stability and viral suppression for people living with HIV/AIDS: Findings from the Building a Medical Home for Multiply Diagnosed HIV positive Homeless Populations Initiative

PONE-D-19-29536R2

Dear Dr. Rajabiun,

We’re pleased to inform you that your manuscript has been judged scientifically suitable for publication and will be formally accepted for publication once it meets all outstanding technical requirements.

Kind regards,

Zixin Wang, PhD.

Academic Editor

PLOS ONE
---

## [Editor Report · Acceptance letter]

15 Sep 2020

PONE-D-19-29536R2 

Pathways to housing stability and viral suppression for people living with HIV/AIDS: Findings from the Building a Medical Home for Multiply Diagnosed HIV positive Homeless Populations Initiative 

Dear Dr. Rajabiun:

I'm pleased to inform you that your manuscript has been deemed suitable for publication in PLOS ONE. Congratulations! Your manuscript is now with our production department. 

Kind regards, 

on behalf of

Professor Zixin Wang 

Academic Editor

PLOS ONE